# Increased Ruminoreticular Temperature and Body Activity after Foot-and-Mouth Vaccination in Pregnant Hanwoo (*Bos taurus coreanae*) Cows

**DOI:** 10.3390/vaccines9111227

**Published:** 2021-10-22

**Authors:** Daehyun Kim, Jaejung Ha, Joonho Moon, Doyoon Kim, Wonhee Lee, Chanwoo Lee, Danil Kim, Junkoo Yi

**Affiliations:** 1Livestock Research Institute, Gyeongsangbuk-Do, 186 Daeryongsan-ro, Anjeong-myeon, Yeongju 36052, Korea; chunja2411@korea.kr (D.K.); hjjggo@korea.kr (J.H.); 2Lartbio Co., Ltd., 12th Floor, 234 Teheran-ro, Gangnam-gu, Seoul 06221, Korea; kuma618@gmail.com; 3Department of Animal Science and Biotechnology, Kyungpook National University, Sangju 41566, Korea; kdy51311@naver.com; 4Department of Biotechnology, College of Agriculture & Life Science, Hankyong National University, Ansung 17579, Korea; wonhee0625@naver.com (W.L.); rhdaud2299@naver.com (C.L.); 5Institute of Green-Bio Science and Technology, Seoul National University, Seoul 25354, Korea

**Keywords:** ruminal biocapsule sensor, FMD vaccination, ruminoreticular temperature, body activity, abortion, premature birth, Korean cattle

## Abstract

How does vaccination against foot-and-mouth disease (FMD) affect pregnant cows? Vaccination is the most effective method of preventing the spread of FMD, but it is linked to sporadic side effects, such as abortion and premature birth, which result in economic loss. In this study, ruminoreticular temperature and body activity were measured before and after FMD vaccination using a ruminoreticular biocapsule sensor in Hanwoo cows at different stages of pregnancy. Compared to the unvaccinated groups, the ruminoreticular temperature increased 12 h after vaccination in the vaccinated groups. This increase in temperature is significantly correlated to vaccination. Compared to the nonpregnant and early pregnancy groups, the ruminoreticular temperature of the late pregnancy group increased sharply by more than 40 °C. Moreover, in nonpregnant and early pregnancy groups, a rapid increase in body activity was observed after FMD vaccinations. Of the 73 pregnant vaccinated cows in the study, a total of five cases had side effects (four abortions and one premature birth). Therefore, changes in the ruminoreticular temperature and activity in pregnant cows can be used as raw data to further clarify the association of FMD vaccination with the loss of a fetus and possibly predict abortion, miscarriage, and premature birth following FMD vaccination.

## 1. Introduction

Foot-and-mouth disease (FMD) is an infectious disease that causes serious economic losses in the cattle industry and occurs in many countries in North America, Western Europe, South America, and Asia [1]. As vaccination is the most effective way to prevent FMD, many countries have made it obligatory [2,3]. Various types of FMD vaccines, including adeno- and viral vector-based vaccines, virus-like particles, peptide and DNA vaccines, plant-based vaccines, and inactivated vaccines, have been developed. The inactivated FMD vaccine is the most commonly used [4,5,6,7,8].

Unfortunately, there are reports of unfavorable side effects following FMD vaccination in cattle, e.g., early embryo loss, low sperm fertility, decreased milk production, increased acute-phase reaction and anovulation, and increased ruminal temperature in nonpregnant cows [8,9,10,11]. In addition, it has been reported that the level of inflammatory cytokines and gene expression can increase due to the acute-phase immune response and that the innate immune response can be induced by the oil-based adjuvant in the FMD vaccine [3,8,12,13]. FMD vaccination 30 days after artificial insemination has been reported to increase rectal temperature, early embryo loss, and haptoglobin levels in the blood of cattle. Increases in temperature and haptoglobin levels are highly correlated with an acute-phase immune response [8,10].

The most common clinical symptoms of the FMD virus in Korean cattle are increased body temperature, decreased appetite, blisters (in the mouth, tongue, hooves, and nipples), and miscarriage [14]. In previous studies, we confirmed the increased ruminoreticular temperature and acute immune response after FMD vaccination in nonpregnant cows [11]. However, there are no reports on changes in ruminoreticular temperature and body activity following FMD vaccination or on unfavorable side effects associated with pregnancy, such as abortion or premature birth. Therefore, this study aims to examine the differences in ruminoreticular temperature and body activity relative to pregnancy stage after FMD vaccination in Hanwoo cows.

## 2. Materials and Methods

### 2.1. Animals and Management

This study examined 203 cows (63 cows pregnant for over 210 days bred in a space of 10 m^2^/cow, 72 cows pregnant for over 80 days, and 68 nonpregnant cows both bred in a space of 20 m^2^/cow) equipped with a ruminoreticular biocapsule sensor and reared in the same farm (Gyeongsangbuk-Do Livestock Research Institute). Detailed information on the age of the cows, days of pregnancy, and age at vaccination in the experimental groups is shown in Table 1. Cows were fed according to the Korean Cattle Feeding Management Regulations and kept in a breeding space conforming to legal requirements (>10 m^2^/cow), with a stanchion installed. All experiments were conducted with the approval of the Animal Ethics Committee of the Gyeongsangbuk-do Livestock Research Institute (Approval number: #106).

### 2.2. Real-Time Measurement of Ruminoreticular Temperature and Body Activity

The ruminoreticular temperature and body activity were measured with the biocapsule sensor for 30 days after vaccination. Information on the biocapsule sensor (LiveCare; uLikeKorea, Seoul, Korea) used to measure ruminoreticular temperature and measurement of body activity can be found in the work of Kim and Choi et al. [15,16,17]. Body activity (V) is expressed as the root value of the sum of X^2^ + Y^2^ + Z^2^ measured with an indwelling 3-axis accelerometer (X, Y, and Z). Detailed information on this technique has been reported [17].

### 2.3. FMD Vaccination

The FMD vaccine (Daesung microbiological labs co., FMD vaccine, O manisa, O3039, A22 Iraq, Seoul, Korea) was administered intramuscularly with Anti-VS-2 Inj. (Samyang Anipharm Co., Pocheon, Korea) for vaccine stress relief. The same dose of FMD vaccine from the same batch was injected intramuscularly into the necks of the cattle.

### 2.4. Pregnancy Test

The pregnancy state of the cattle was evaluated through rectal ultrasonography (HONDA HS-101V, HONDA Co., Ltd., Tokyo, Japan) 10 days before the initiation of the experiment. An additional pregnancy test was performed to determine whether miscarriage had occurred 10 days after FMD vaccination.

### 2.5. Statistical Analysis

The PRISM program (version: 8.1.0) was used for statistical analysis. Average ruminoreticular temperature and body activity before and after FMD vaccination were compared for each of the 3 groups through two-way ANOVA (Tukey’s multiple comparison test) for each 6 h period. The significance level was set at * *p* < 0.05, ** *p* < 0.01, and *** *p* < 0.001.

## 3. Results

### 3.1. Change of Ruminoreticular Temperature and Body Activity following FMD Vaccination

For nonpregnant cows (NP), the vaccinated group is hereinafter referred to as “FMD_NP”, and the unvaccinated group as “Control_NP” (Figure 1A and Figure 2A and Table 1). For cows pregnant for less than 80 days, the vaccinated group is hereinafter referred to as “FMD < P80” and the unvaccinated group as “Control < P80” (Figure 1B and Figure 2B and Table 1). For cows pregnant for more than 210 days, the vaccinated group is hereinafter referred to as “FMD > P210”, and the unvaccinated group as “Control > P210” (Figure 1C and Figure 2C and Table 1).

The ruminoreticular temperature of all the test groups gradually increased starting at 12 h after the vaccination (Figure 1, *p* < 0.005). The ruminoreticular temperature of the FMD_NP group 12 h after vaccination (39.1 ± 0.09 °C) was higher than that the unvaccinated Control_NP group (38.6 ± 0.00 °C) remained at (Figure 1A). The ruminoreticular temperature of the FMD < P80 group 12 h after vaccination (39.3 ± 0.09 °C) was higher than that of the Control < P80 group (38.5 ± 0.05 °C) (Figure 1B). The ruminoreticular temperature in the FMD > P210 group 12 h after vaccination (39.9 ± 0.18 °C) was significantly higher than the Control > P210 group (39.0 ± 0.06 °C) (Figure 1C).

At the time of FMD vaccination, a rapid increase in body activity was observed in the FMD_NP (1846.7 ± 133.7 V) and FMD < P80 groups (2156.1 ± 117.7 V) but did not change in the FMD > P210 group (1251.5 ± 41.3 V) (Figure 2). The body activity of the FMD_NP group increased (1846.7 ± 133.7 V) but did not change in the Control_NP group (1251.1 ± 13.2 V) (Figure 2A). At the time of FMD vaccination, the body activity of the FMD < P80 group was higher (2156.1 ± 117.7 V) than that of the Control < P80 group (1232.6 ± 6.2 V) (Figure 2B). There was no difference when comparing the body activity at the time of FMD vaccination in the FMD > P210 group (1251.5 ± 41.3 V) and the Control > P210 group (1229.4 ± 6.8 V) (Figure 2C).

### 3.2. Cases of Abortion and Premature Birth following FMD Vaccination

In this study, after FMD vaccination, a total of 5 (6.8%) out of 73 pregnant cows experienced abortions and premature births, with four abortions and one premature birth identified by clinical criteria. There were no abortions or premature deliveries in unvaccinated cows. The death of the aborted or premature calves was confirmed visually and measurements were taken to determine the fetal length and weight.

In premature birth #13–81, the cow was vaccinated on the 274th day of pregnancy. It showed a rapid increase in ruminoreticular temperature 12 h after vaccination. The birth weight of the calf was 22 kg (Appendix A).

In case #433, the cow was vaccinated on the 267th day of pregnancy. It showed a low ruminoreticular temperature immediately after vaccination and delivered 26 h after vaccination. However, the calf was born 18 days early and died immediately after delivery. Its birth weight was 20 kg (Appendix A).

In case #17–25, the cow was vaccinated on the 210th day of pregnancy. It showed a high ruminoreticular temperature of 40 °C 18 h after vaccination, and the fetus was aborted 4 days after vaccination. The aborted fetus was 65 cm long and weighed 13 kg (Appendix A).

In case #425, the cow was vaccinated on the 68th day of pregnancy. It experienced an abortion 4 days after vaccination, and the aborted fetus died. The occurrence of miscarriage was immediately reconfirmed through a rectal examination using ultrasound equipment, but the aborted fetus could not be visually confirmed (Appendix A).

In case #398, the cow was vaccinated on the 47th day of pregnancy. It showed a high ruminoreticular temperature of 40 °C at 12 h after vaccination and experienced an abortion 5 days after vaccination (Appendix A). Although the cow was examined through ultrasound equipment, the fetus could not be visually confirmed, as in case #425.

## 4. Discussion

The association of ruminoreticular temperature and body activity with FMD vaccination was analyzed in Korean cattle. After vaccination, the ruminoreticular temperature rose between 12 to 30 h, and body activity transiently increased at the time of vaccination. The correlation between a rise in ruminoreticular temperature and vaccination was statistically significant. A total of 203 cows (63 cows pregnant for over 210 days, 72 cows pregnant for over 80 days, and 68 nonpregnant cows) were tested for adverse effects following FMD vaccination, so the ruminoreticular temperature and body activity of pregnant cows could be examined in detail. In unvaccinated controls, the body temperature could conceivably rise temporarily due to a placebo effect. However, a previous study, in which 0.9% normal saline was injected instead of the FMD vaccine, obtained similar results, supporting the lack of a placebo effect [11]. Therefore, the results of this study are likely not due to a placebo effect.

A total of five cows experienced abortions and premature births following FMD vaccination. There were no changes in the external environment (movement of space, cleanliness, construction, etc.) before and after the test period. Therefore, these events are most likely due to the vaccination.

Studies have reported that cow parity, breeding season, insemination protocol, and calving interval affect the incidence of pregnancy loss [18]. In addition, it has been reported that when artificial insemination is performed 30 days after FMD vaccination early embryo loss were more frequent. On the meantime authors reported increases in rectal temperature and in acute immune response protein (haptoglobin) in the blood [8]. Therefore, we hypothesize that the cause of miscarriage and premature birth is an acute immune response to FMD vaccination, but the exact reason is still unclear. Additionally, methods for validating abortions and preterm births due to FMD vaccination are not yet established. It is generally accepted that the main causes of miscarriage in cattle are heat stress, season, milk production, high environmental temperature, vaccination, and bacterial infection [18,19,20]. The general incidence of such miscarriages has been reported to be from 2 to 5% and to occur most frequently at 30–60 days of gestation [21,22,23]. The average prevalence of miscarriage in this study was 6.8% among 73 pregnant cows at the time of vaccination. It was not possible to compare this statistically to previously reported prevalence estimates due to the small number of cases and lack of an appropriate comparison group.

The cases of abortion on the 47th and 68th days of gestation could be considered as embryo losses. However, we classified them as abortions in this study because there were no other environmental changes other than the FMD vaccination. In the case of abortion or premature birth after FMD vaccination, numerous tests for bacteria and viruses are required to scientifically demonstrate the cause and mechanism. Once the role of bacteria and viruses is disconfirmed using the reported methods, the effect of FMD vaccination on abortion can be estimated, but the exact result cannot be known. A method that can completely verify abortion, premature birth, and early embryo loss due to FMD vaccination has not yet been developed.

In addition, as shown in Figure 1, the temporary decrease in ruminoreticular temperature occurred at similar times every day. The main cause of this phenomenon is the intake of drinking water accompanied by feed intake, and because the biosensor is in the rumen and reticulum, a temporary drop in temperature can be seen. Similar results due to water consumption have been reported elsewhere [24,25].

In the case of pregnancies longer than 210 days, it is possible to confirm that the temporal ruminoreticular temperature pattern is different because those cows were fed after 6 h (fed at 4:00 PM), unlike cows that were pregnant for less than 80 days and nonpregnant cows (fed at 10 AM), which is carried out to induce birthing at sunrise (Figure 1C).

Additional findings in this study were that after FMD vaccination, relative to the Control > P210 group, the ruminoreticular temperature increased the most in the FMD > P210 group, but there was no difference in body activity. We postulate that the reason for this lack of difference in body activity among cows that are pregnant for more than 210 days is because nonpregnant cows and cows pregnant for less than 80 days are bred in a larger space of 300 m^2^/15 cows (20 m^2^/cow), whereas cows with over 210 days of pregnancy are bred in a space of only 20 m^2^/2 cows (10 m^2^/cow) to avoid damage to the calves. Therefore, the likely lack of difference in activity in cows at over 210 days of pregnancy is due to their narrow breeding space.

## 5. Conclusions

In summary, data on changes in ruminoreticular temperature and body activity can be used as raw data for the further correlation of abortion and premature birth to FMD vaccination. Rises in ruminoreticular temperature are particularly useful due to their strong correlation with vaccination. This will allow for the prediction of abortion and premature birth after vaccination and help in drug development to prevent adverse effects. These findings could ultimately lead to the development of an artificial intelligence system that can manage these predictions.

## Figures and Tables

**Figure 1 vaccines-09-01227-f001:**
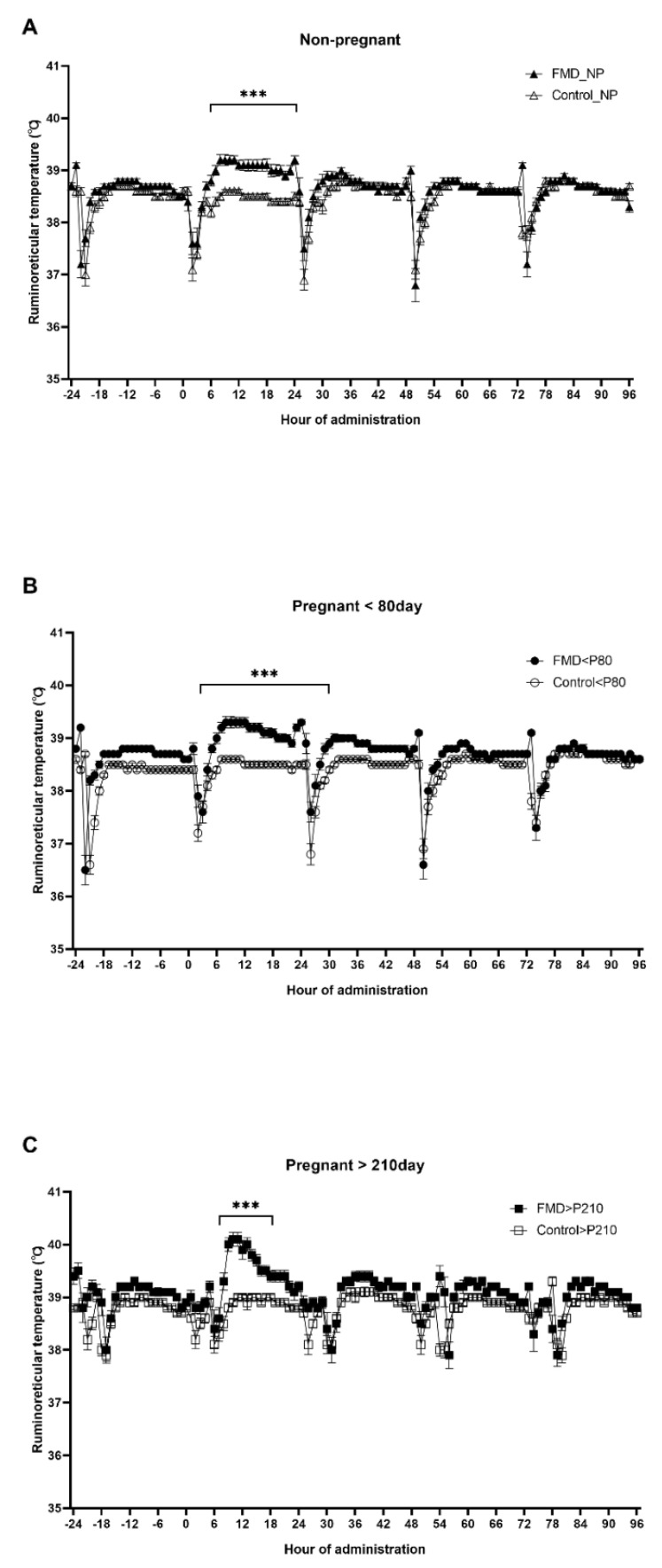
Change in ruminoreticular temperature depending on pregnancy and FMD vaccination (n = 203). (**A**) Ruminoreticular temperature of control and FMD−vaccinated nonpregnant cows. The black line connected by “△” represents the mean temperature of the Control_NP group (not vaccinated). The black line connected by “▲” represents the mean temperature of the FMD_NP group (vaccinated). (**B**) Ruminoreticular temperature of control and FMD-vaccinated cows pregnant for less than 80 days. The black line connected by “○” represents the mean temperature of the Control < P80 group (not vaccinated). The black line connected by “●” represents the mean temperature of the FMD < P80 group (vaccinated). (**C**) Ruminoreticular temperature of control and FMD−vaccinated cows pregnant for more than 210 days. The black line connected by “□” represents the mean temperature of the Control > P210 group (not vaccinated). The black line connected by “■” represents the mean temperature of the FMD > P210 group (vaccinated). Day 0 is the time of the FMD vaccination and the error bar is the standard error of the mean. *** means significance level *p* < 0.001.

**Figure 2 vaccines-09-01227-f002:**
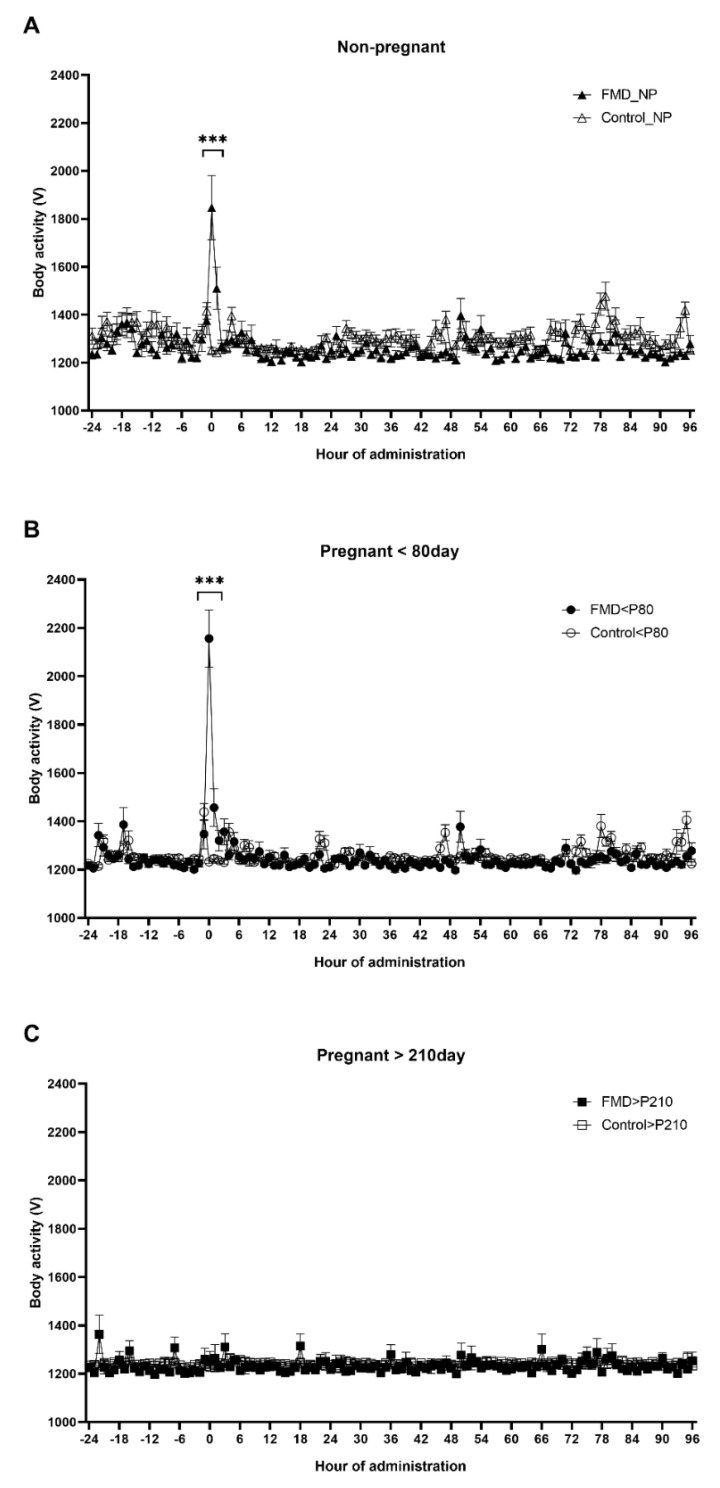
Change in body activity depending on pregnancy and FMD vaccination (n = 203). (**A**) Body activity of control and FMD−vaccinated nonpregnant cows. The black line connected by “△” represents the mean movement of the Control_NP group (not vaccinated). The black line connected by “▲” represents the mean movement of the FMD_NP group (vaccinated). (**B**) Body activity of control and FMD-vaccinated groups of cows pregnant for less than 80 days. The black line connected by “○” represents the mean movement of the Control < P80 group (not vaccinated). The black line connected by “●” represents the mean movement of the FMD < P80 group (vaccinated). (**C**) Body activity of control and FMD vaccination group in cows pregnant for over 120 days. The black line connected by “□” represents the mean movement of the Control > P210 group (not vaccinated). The black line connected by “■” represents the mean movement of the FMD > P210 group (vaccinated). Day 0 is the time of the FMD vaccination and error bar is the standard error of the mean. *** means significance level *p* < 0.001.

**Table 1 vaccines-09-01227-t001:** Information for experimental groups (n = 203). NP, nonpregnant; <P80, pregnant for less than 80 days; >P120, pregnant for over 120 days.

Group		n	Months of Age	Days of Pregnancy	Breeding Space
Control	NP	29	44.3 ± 2.9	0	300 m^2^/15 cows
	<P80	28	42.7 ± 2.0	57.5 ± 3.0	
	>P210	34	50.4 ± 2.7	258.6 ± 2.7	20 m^2^/2 cows
FMD Vaccinated	NP	39	34.3 ± 2.4	0	300 m^2^/15 cows
	<P80	44	38.5 ± 1.6	49.9 ± 2.3	
	>P210	29	42.5 ± 3.4	256.9 ± 3.2	20 m^2^/2 cows
Total		203	42.1 ± 3.8	-	-

## Data Availability

Not applicable.

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
