# Peer review of "Increased Ruminoreticular Temperature and Body Activity after Foot-and-Mouth Vaccination in Pregnant Hanwoo (Bos taurus coreanae) Cows"

_vaccines, 2021, doi:10.3390/vaccines9111227_

Round 1
Reviewer 1 Report
The case report by Kim et al. analyses safety of a commercial Foot and Mouth Disease (FMD) vaccine in pregnant Hanwoo cattle, revealing an association between vaccination and negative pregnancy outcomes in cows at an advanced stage of gestation. The study appears well designed and the data are convincing to suggest a link between abortion or premature birth in a minority of cows at advanced stages of pregnancy. I do however have a few questions that should be addressed:
What was the rate (if any) of abortion or premature delivery in the non-vaccinated control group? We presume this is low or even zero, but I can’t see it explicitly mentioned.
While heat stress is mentioned as a potential mechanism underlying spontaneous abortion, and links to elevated body temperature post-vaccination are clear, what is the link between reduced body activity? This is not clear to a non-expert.
Minor:
Page 1, line 23, alter “significantly” to ‘significant’
Author Response
The Review Report is attached below.

Reviewer 2 Report
This is an interesting study. The study indicates an association between vaccination and a transient rise in body temperatures of the cattle. There are a few issues to be addressed:
1) Were the non-vaccinated cows injected with a placebo? If not, it is possible that the responses were associated with the stress of the injection, rather than the vaccine. This should be clarified in the methods, and if the placebo was not given, there should be a discussion point made about this.
2) The initial focus in the discussion is about the abortions that occurred in the vaccinated cattle. This is really a side result of this study, which the discussion indicates that these results are supportive but not definitive. In fact, there is no indication whether there were any abortions occurring in the control (non vaccinated) group. If there were none, is the result statistically significant?
3) I would recommend putting the body temp discussion first in that section and then follow with the potential association with abortion.
Author Response
The Review Report is attached below.

Round 2
Reviewer 2 Report
The authors responded well to comments in previous review.
Author Response
"Please see the attachment."
